# Parallel evolution of dominant pistil-side self-incompatibility suppressors in *Arabidopsis*

Sota Fujii [1,2,3,6], Hiroko Shimosato-Asano[2,6], Mitsuru Kakita[2,4], Takashi Kitanishi[2], Megumi Iwano[2,5] & Seiji Takayama [1,2✉]

Selfing is a frequent evolutionary trend in angiosperms, and is a suitable model for studying the recurrent patterns underlying adaptive evolution. Many plants avoid self-fertilization by physiological processes referred to as self-incompatibility (SI). In the Brassicaceae, direct and specific interactions between the male ligand SP11/SCR and the female receptor kinase SRK are required for the SI response. Although *Arabidopsis thaliana* acquired autogamy through loss of these genes, molecular evolution contributed to the spread of self-compatibility alleles requires further investigation. We show here that in this species, dominant *SRK* silencing genes have evolved at least twice. Different inverted repeat sequences were found in the relic *SRK* region of the Col-0 and C24 strains. Both types of inverted repeats suppress the functional *SRK* sequence in a dominant fashion with different target specificities. It is possible that these dominant suppressors of SI contributed to the rapid fixation of self-compatibility in *A. thaliana*.

[1] Graduate School of Agricultural and Life Sciences, The University of Tokyo, Tokyo 113-8657, Japan. [2] Graduate School of Biological Sciences, Nara Institute of Science and Technology, Ikoma 630-0192, Japan. [3] Japan Science and Technology Agency, Precursory Research for Embryonic Science and Technology, Saitama 332-0012, Japan. [4] Present address: Center for Research Administration and Collaboration, Tokushima University, Tokushima 770-8506, Japan. [5] Present address: Graduate School of Biostudies, Kyoto University, Kyoto 606-8502, Japan. [6] These authors contributed equally: Sota Fujii, Hiroko Shimosato-Asano. ✉email: a-taka@mail.ecc.u-tokyo.ac.jp

Self-incompatibility (SI) refers to mechanisms that prevent self-fertilization and promote outcrossing between different individuals adopted by over 40% of flowering plant species[1]. In the Brassicaceae family, SI is controlled by two multi-allelic genes: the *S-locus protein 11* (*SP11*; also known as the *S-locus cysteine-rich protein*, *SCR*) and the *S-locus receptor kinase* (*SRK*)[2,3]. SP11/SCR is a peptide ligand localized to the pollen coat surface[4,5], and SRK is a stigma surface receptor[6]. Proteins translated from a tightly linked pair (haplotype) of *SP11/SCR* and *SRK* directly and specifically bind to one another, triggering a self-pollen rejection signal in the stigma[7].

On the other hand, the evolution of self-compatibility (SC) is one of the most frequently occurred convergent evolution known in plants[8]. In the model plant species *Arabidopsis thaliana*, SC is considered to have evolved at least three times independently because three major haplogroups (A, B, and C) derived from a much larger set of diverged *S* haplotypes in the closely related outcrossers, such as *A. halleri*, are found[9]. A previous study further classified the alleles into 12 haplotypes[10]. Strains such as Col-0, Uk-3 and Wei-1 belong to haplogroup A in which Col-0 carries a pseudo *SRK* and a non-functional *SP11/SCR*, whereas Uk-3 and Wei-1 carry functional *SRK* genes[11]. The inversion of *SP11/SCR* is considered to be the primary mutation disrupting the SI of this haplogroup[11]. The Cvi-0 strain belongs to haplogroup B in which a premature stop codon is found in *SRK*[9]. Strains such as C24 belong to haplotype R2 formed by recombination between the A and C haplogroups, followed by large deletions of both *SRKA* and *SRKC*[10]. Notwithstanding these facts, molecular forces contributed to the spread of SC in this species still requires understanding.

Recently, some laboratory observations that provide support for this question have been reported. Although all known *A. thaliana* strains are self-compatible, SI can be reconstituted in this species when functional forms of *SP11/SCR* and/or *SRK* are introduced[2,11,12]. These findings indicate that at least in some strains, non-*S*-locus components required to express SI are still retained; however, there has been some controversy over the requirements for genetic factors to express the female SI-phenotype[13,14]. When the Col-0 strain was transformed with the *S_b*-haplotype, *SRKb* and *SCRb* genes from *A. lyrata* (an allele corresponding to *SRK*20 (ref. [15]), the transgenic plants showed only weak or transient SI phenotypes[12]. This transient SI phenotype was evident in stigmas at flower stage 13 (at anthesis) but was attenuated in later stages. On the other hand, robust SI was expressed regardless of the flower stage in other *A. thaliana* strains such as C24, when transformed with the same genes. Liu et al. reported that this phenotypic variation is due to natural variation in the *S*-locus-linked *PUB8* gene, which encodes an ARM-repeat U-box-containing protein[16]. In contrast, Indoriolo et al. reported that another ARM-repeat U-box-containing protein, ARC1, is required to express strong SI in the Col-0 and Sha strains[13]. ARC1 is involved in the expression of SI in *Brassica* species[17], and the *ARC1* gene has been independently deleted in many self-compatible species of this genus[18]. The discrepancies among these reports have not yet been resolved. Identifying the prerequisites for reconstituting SI in *A. thaliana* directly relates to solving how SC evolved in this species.

Through the process of identifying the differences between Col-0 and C24 in this study, we found that an inverted repeat sequence (*SRKIR^Col-0*) in the relic *SRKA* region of Col-0 suppresses *SRK* mRNA accumulation by different haplotypes. Interference by the small RNAs possibly produced from *SRKIR^Col-0* can explain the instability of the reconstituted SI phenotype in this strain. Furthermore, we found a distinct inverted repeat in C24 (*SRKIR^C24*) that may also suppress the expression of allelic *SRK* genes with a target specificity different from *SRKIR^Col-0*.

These two sequence structures evolved independently in *A. thaliana*. We discuss the possible contributions of these dominant suppressors in the fixation of SC in *A. thaliana*. Our results may also support the idea that inverted repeats are a recurrent evolutionary form of allelic dominance.

## Results

**The Col-0 strain may carry a dominant suppressor of *SRK* mRNA.** To confirm the instability of the SI phenotype in Col-0, we generated a transgenic line carrying *SRKb* and *SCRb* that were cloned from the *A. lyrata* $S_b$ ($S_{20}$) haplotype (Col-0 + *SRKb-SCRb*). In a previous study, we produced a C24 transgenic line carrying single copies of *SRKb* and *SCRb* (called SI-C24 in this study for brevity)[19]. SI-C24 exhibited SI at both flower stages 13 and 14 (Supplementary Fig. 1). Col-0 + *SRKb-SCRb* pistils at stage 13 were only partially able to reject pollen from SI-C24 flowers, and at stage 14 they failed to reject (Supplementary Fig. 1). These observations were in agreement with the previous results of Nasrallah et al.[12].

When Col-0 + *SRKb-SCRb* was used as the pollen donor with SI-C24 pistils from flower stage 14, virtually no pollen tube growth was observed (Fig. 1a, b). Therefore, Col-0 was able to express the *SCRb* gene and show the male side of the SI phenotype but was unable to express the female side of the SI phenotype. The $F_1$ generation of SI-C24 and Col-0 failed to express a strong female SI phenotype at stage 14 (Fig. 1b) and the *SRKb* mRNA levels were significantly reduced (Fig. 1c). We, therefore, hypothesized that Col-0 carries a dominant endogenous factor, not present in C24, that suppresses the expression of *SRKb*. This was not in agreement with the previous study by Liu et al. reported that the weak SI expression phenotype is recessive[16].

**Small RNAs are possibly produced from *SRKIR^Col-0*.** Col-0 carries a relic *S* locus sequence belonging to the A haplogroup (At4g21370: *ψSRKA*). From the available genomic information archived in The Arabidopsis Information Resource (TAIR), we found an inverted repeat structure within the Col-0 allele of *ψSRKA* and named this structure *SRKIR^Col-0*. *SRKIR^Col-0* consists of a 502-bp inverted repeat, which corresponds to the fragment encoding the SRK kinase domain in the orthologous *SRK* sequence from the *A. halleri* $S_4$ haplotype (Fig. 1d). Such a hairpin RNA sequence can form a duplex-RNA and could be processed into small RNAs by Dicers[20]. We suspected that *SRKIR^Col-0* serves as the source of small RNAs that act to suppress expression of the introduced *SRKb*. Consistent with this hypothesis, a transcriptome analysis revealed that 25,671 small RNA reads from Col-0 stigmas mapped to the *SRKIR^Col-0* region (Fig. 1e, Supplementary Fig. 2a). Over 99.5% of the reads (25,563) were uniquely mapped to the *SRKIR^Col-0* region; thus, we considered that these small RNAs were likely produced from this hairpin. The majority of these small RNAs had lengths of 20 or 21 nucleotides (Supplementary Fig. 2b). *SRKIR^Col-0* showed high sequence homology with the DNA sequence encoding the SRK kinase domain in *SRKb*. In particular, the fragment between base-pairs 102 to 253 of the *SRKIR^Col-0* sequence was 89.0% identical to the corresponding region of the *SRKb* gene (Supplementary Fig. 3). We used the psRNAtarget program[21] to predict if these small RNAs that mapped to *SRKIR^Col-0* can target *SRKb*. As a result, 58 non-redundant reads perfectly matched (expectation = 0 in the psRNAtarget program) within the kinase domain of *SRKb* (Supplementary Fig. 4), suggesting that these small RNAs found within the *SRKIR^Col-0* sequence may potentially target *SRKb*.

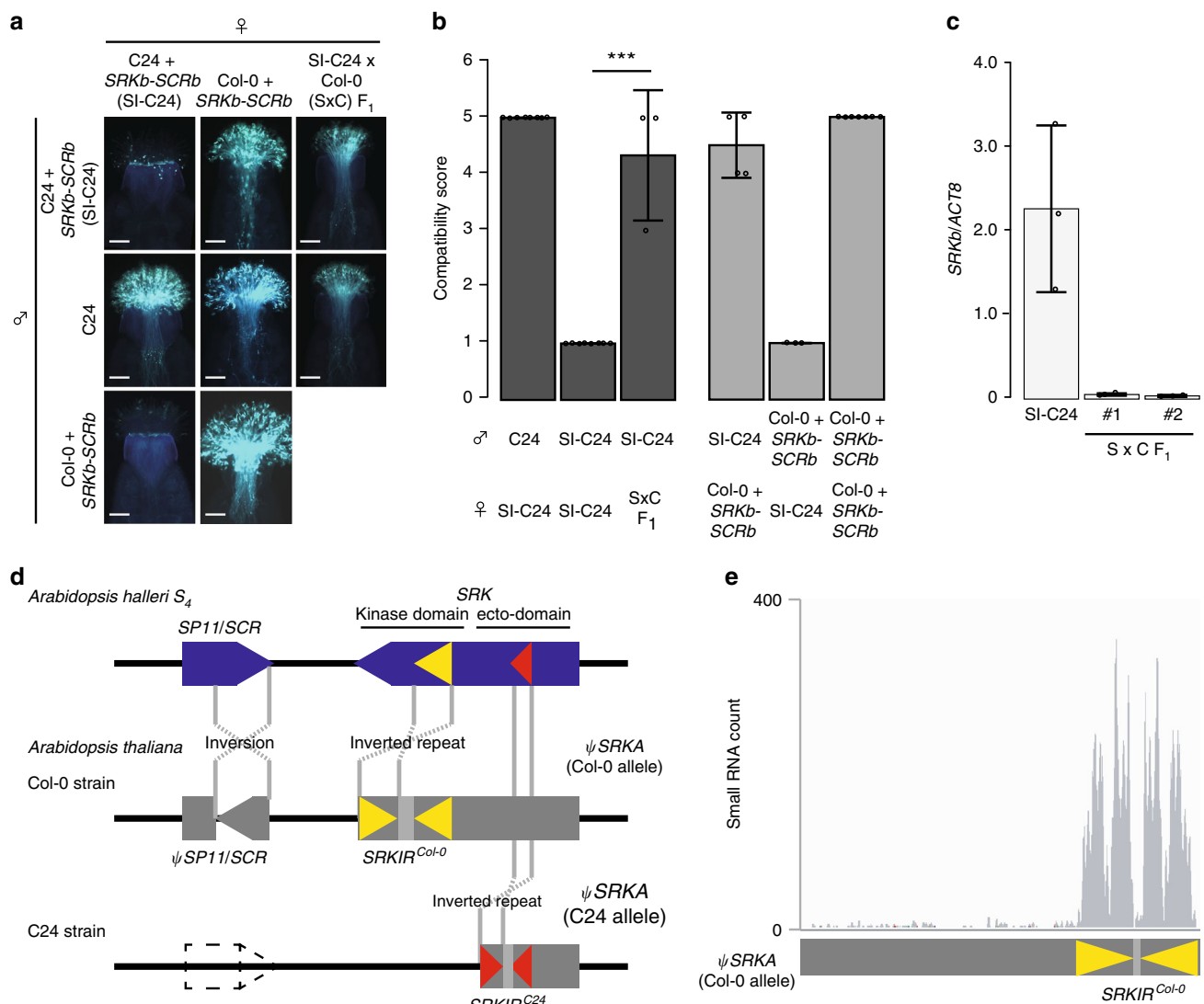

**Fig. 1 *SRKb* mRNA expression in the Col-0 × SI-C24 F$_1$ hybrids and accumulation of small RNAs derived from *SRKIR*$^{Col-0}$. a** Representative fluorescent images of aniline blue-stained pistils at stage 14 three hours after pollination. Scale bars = 100 μm. **b** Summary of pollination tests. S × C F$_1$ is the hybrid obtained from the cross: ♀SI-C24 × ♂Col-0. Compatibility phenotypes were scored as described in the methods. Statistically significant differences based on Dunnett's test are indicated by ***($p < 0.005$), with the exact $p$-value being 7.16e−06. Error bars indicate standard deviations. The number of pistils pollinated (from left to right): $n = 8$, $n = 8$, $n = 8$, $n = 4$, $n = 3$, $n = 7$. **c** Relative mRNA accumulation levels of *SRKb* compared to *ACTIN8* (*ACT8*) in stigmatic tissues of the indicated lines. Two independent S × C F$_1$ individuals (#1 and #2) were analyzed. Error bars indicate standard deviations from independent RNA extraction trials ($n = 3$). **d** Illustration comparing the *S* loci of *A. thaliana* Col-0 and C24, and the *A. halleri S$_4$* haplotype. Col-0 carries an inverted repeat with homology to the sequence encoding the kinase domain of SRK. The C24 strain carries an inverted repeat with homology to the sequence encoding the ecto-domain of SRK. **e** A histogram showing the numbers of small RNAs from stigmas that map to the pseudogenized *SRKA* gene (ψ*SRKA*) of Col-0; most small RNAs map to the inverted repeat region (see Supplementary Fig. 2a for the mapping results for the flanking regions).

***SRKIR*$^{Col-0}$ suppresses *SRKb* expression in a dominant fashion**. To show whether *SRKIR*$^{Col-0}$ could suppress *SRKb*, we introduced a 5,353-bp genomic fragment containing the *SRKIR*$^{Col-0}$ sequence (full *SRKIR*$^{Col-0}$) into the SI-C24 line (Fig. 2a). In a previous study, we showed that the promoter of ψ*SRKA* is actively transcribed in stigmas[19]. As a result, the SI phenotype observed in the SI-C24 line at stage 14 was compromised in the transformants (Fig. 2b–d). This observation further indicated that the *SRKIR*$^{Col-0}$ sequence can suppress *SRKb* in a dominant fashion. When we introduced a partial fragment that contains only the first repeat of the *SRKIR*$^{Col-0}$ sequence (partial *SRKIR*$^{Col-0}$), this fragment was unable to suppress the function of *SRKb* (Fig. 2b–d).

We also investigated the effect of a T-DNA insertion in *SRKIR*$^{Col-0}$. Compared to the individual that did not carry the

T-DNA insertion (−/−), *SRKb* mRNA accumulation was recovered in the line carrying the T-DNA insertion homozygously (+/+) (Supplementary Fig 5b). These results suggested that the continuous inverted repeat is probably required for *SRKIR*$^{Col-0}$ to function in compromising SI. Taken together, we showed that full *SRKIR*$^{Col-0}$ is necessary and is likely to be sufficient to cause breakdown of SI in the C24 background.

***SRKIR*$^{Col-0}$ suppresses the expression of endogenous *SRK*s**. We next tested whether the *SRKIR*$^{Col-0}$ sequence could suppress expression of the endogenous *SRKb* gene. To achieve this objective, we generated inter-specific hybrids of *A. thaliana* and *A. lyrata*. We crossed *A. thaliana* Col-0 and C24 with an individual of *A. lyrata* carrying the *S$_a$* (intermediate dominance class[22]) and

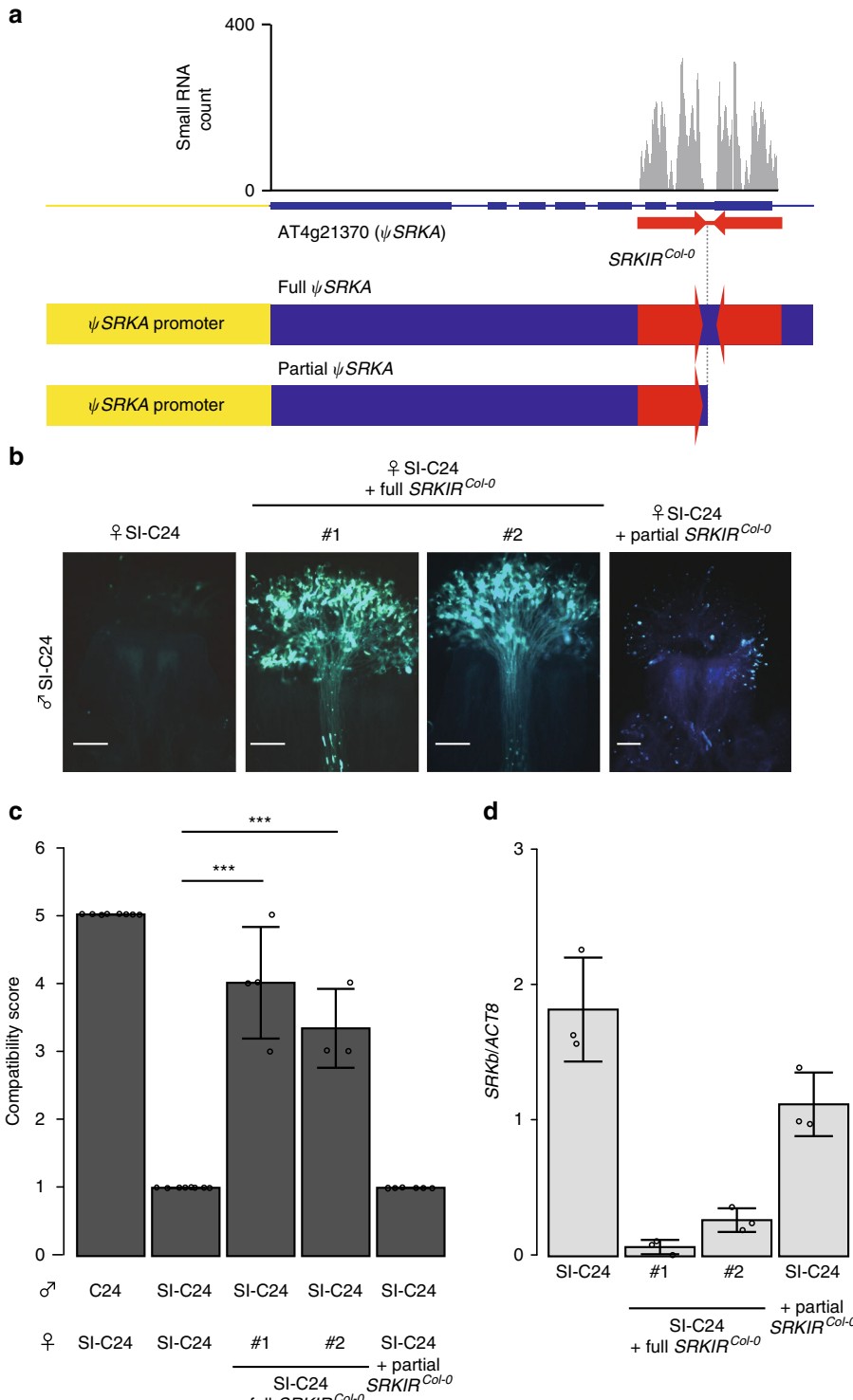

**Fig. 2. Introduction of *SRKIR^{Col-0}* into the SI-C24 line breaks down SI. a** Schematic drawing of the *ψSRKA* locus of Col-0. **b** Representative fluorescent images of aniline blue-stained pistils at stage 14 three hours after pollination. #1 and #2 indicate pistils from two independent first generation transformants. Scale bars = 100 μm. **c** Summary of pollination tests. Statistically significant differences based on Dunnett's test are indicated by ***($p <$ 0.005). The exact $p$-values found by Dunnett's test compared to the ♀SI-C24 × ♂SI-C24 cross were 4.66e−07 and 1.84e−05, from left to right. Error bars indicate standard deviations. The number of pistils pollinated (from left to right): $n = 8$, $n = 8$, $n = 4$, $n = 3$, $n = 6$. **d** Relative mRNA accumulation levels of *SRKb* compared to *ACT8* in stigmatic tissues of the indicated lines. Error bars indicate standard deviations from independent RNA extraction trials ($n = 3$).

$S_b$ (most dominant class[22]) haplotypes. Pistils carrying the $S_b$ allele from the *A. lyrata* parent and the C24 genotype from the *A. thaliana* parent were incompatible with pollen from the $S_aS_b$ *A. lyrata* individual (Fig. 3a,b). It should be noted that pollen from

$S_aS_b$ *A. lyrata* shows only $S_b$ specificity because $S_b$ is dominant over $S_a$ for *SCR* expression[23]. In contrast, pistils with the $S_b$ allele from *A. lyrata* and the Col-0 genotype from *A. thaliana* were compatible with $S_aS_b$ pollen (Fig. 3a, b). We also found that

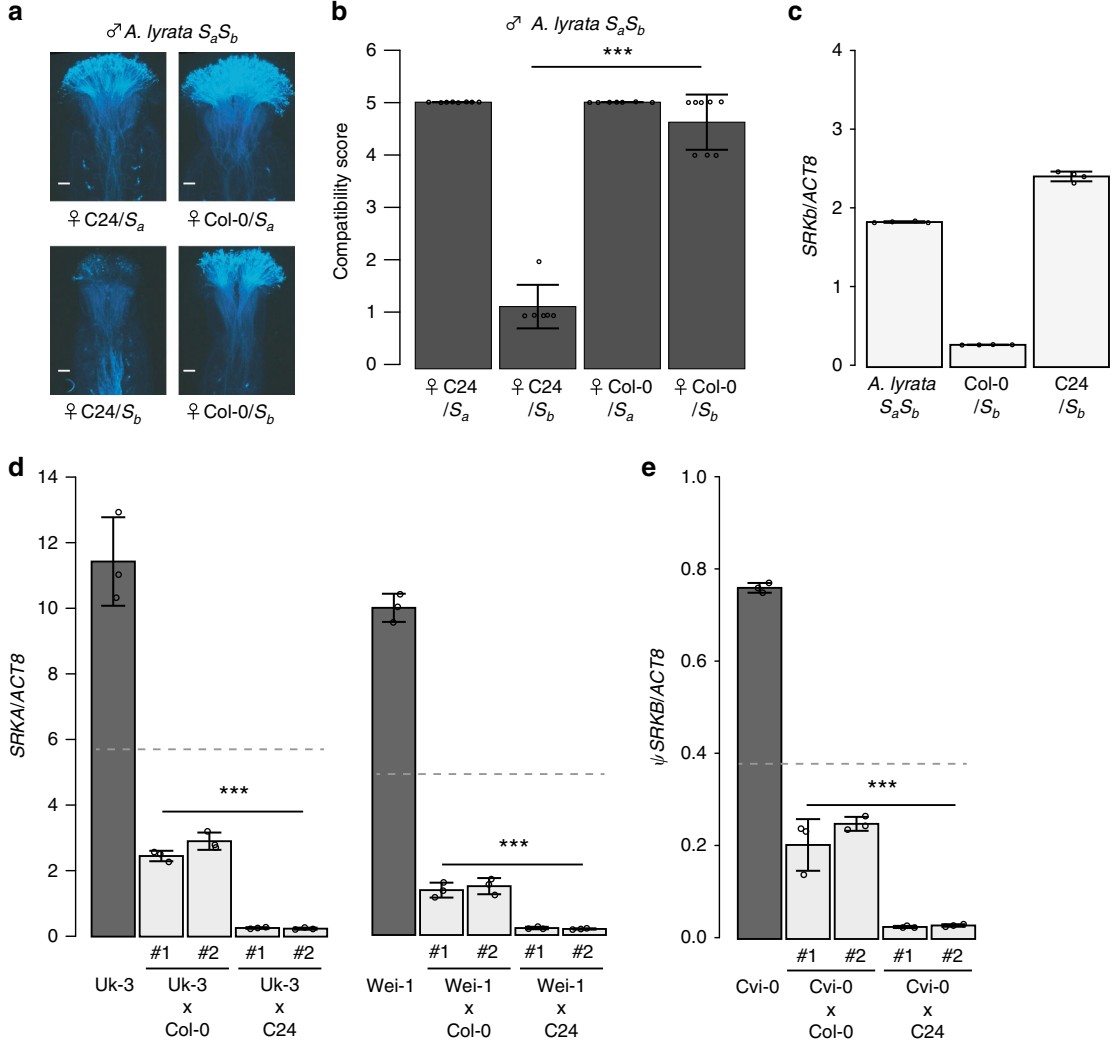

**Fig. 3. $SRKIR^{Col-0}$ and $SRKIR^{C24}$ may suppress multiple endogenous $SRK$ alleles. a** Representative fluorescent images of aniline blue-stained pistils at stage 14 of *A. thaliana–lyrata* inter-species hybrids with different genotypes 3 h after pollination. $C24/S_a$ and $C24/S_b$ are individuals obtained from the cross *A. lyrata* $S_aS_b$ × C24 that inherited the $S_a$ and $S_b$ alleles, respectively. Similarly, $Col-0/S_a$ and $Col-0/S_b$ are *A. lyrata* $S_aS_b$ x Col-0 hybrids that inherited the $S_a$ and $S_b$ alleles, respectively. Pollen from an *A. lyrata* $S_aS_b$ individual was used for the pollination test. Scale bars = 100 μm. **b** Summary of pollination tests. A statistically significant difference between $C24/S_b$ and $Col-0/S_b$ (***$p$ < 0.005) was detected using Dunnett's test, with the exact $p$-value being 0.00e+00. Error bars indicate standard deviations. Number of pistils pollinated (from left to right): $n = 8$, $n = 6$, $n = 7$, $n = 8$. **c** Relative mRNA accumulation levels of $SRKb$ compared to the *A. lyrata ACTIN8* ortholog in stigmatic tissues of the indicated lines. Error bars indicate standard deviations from independent RNA extraction trials ($n = 4$). **d**, **e** Relative mRNA accumulation levels of the $SRK$ genes compared to $ACT8$ in stigmatic tissues of the indicated lines. For each hybrid cross, two individuals (#1 and #2) were tested. In **c**–**e** error bars indicate standard deviations from independent RNA extraction trials ($n = 3$). The broken lines indicate the expected mRNA accumulation level of the $SRK$ genes in the hybrids compared to that of the parents (theoretically one-half). Dunnett's tests were used to find significant differences (***$p$ < 0.005) compared to the expected mRNA level. **d** The exact $p$-values from left to right: 5.02e−13, 2.22e−16, 0.00e+00, 0.00e+00, 1.17e−06, 1.05e−04, 7.71e−12, and 8.88e−16. **e** The exact $p$ values from left to right; 1.63e−05, 9.15e−05, 2.00e−15, and 7.77e−16.

the levels of $SRKb$ mRNA were reduced by 6.5-fold in the $Col-0/S_b$ stigmas when compared with the *A. lyrata* $S_aS_b$ stigmas, and this was not the case with the $C24/S_b$ stigmas (Fig. 3c). Therefore, $SRKb$ expression was suppressed in the stigmas with the Col-0 genetic background.

In some *A. thaliana* strains, including Uk-3 and Wei-1, SRK function is retained[11]; both strains carry an intact $SRK$ gene that belongs to haplogroup A ($SRKA$), and Wei-1 can reject pollen from *A.halleri* carrying $SCRA$[11]. The $SRK$ sequence in the Cvi-0 strain belongs to the B haplotype and is a pseudogene ($\psi SRKB^{Cvi-0}$) but is transcribed in flowers[9]. We expected that expression of the $SRK$ genes from these strains would be suppressed in the presence of $SRKIR^{Col-0}$ because of their high sequence identities

with $SRKIR^{Col-0}$ (Supplementary Fig. 6). The psRNAtarget program also predicted that 1,023 and 91 non-redundant small RNA reads found within the $SRKIR^{Col-0}$ sequence matched perfectly within the kinase domains of $SRKA^{Wei-1}$ and $\psi SRKB^{Cvi-0}$, respectively (Supplementary Fig. 4). To verify this finding, we crossed these strains with Col-0 and analyzed the $SRK$ mRNA levels in the $F_1$ generations. Similar to the SI-C24 x Col-0 $F_1$ hybrids (Fig. 1c), $SRK$ expression in the Uk-3 x Col-0, Wei-1 x Col-0, and Cvi-0 x Col-0 $F_1$ hybrids was significantly reduced when compared with the expected levels of transcript accumulation in the hybrids (Fig. 3d, e). This result indicated a broad function of $SRKIR^{Col-0}$ in $SRK$ suppression.

**A putative inverted repeat structure in *S* locus of C24**. Since C24 can successfully express externally introduced *SRKb* (Fig. 1), the reduction in *SRK* mRNA levels in the Uk-3 × C24, Wei-1 × C24, and Cvi-0 × C24 F$_1$ hybrids when compared with the parental Uk-3, Wei-1, and Cvi-0 strains was unexpected (Fig. 3d, e). C24 carries a relic *S* locus belonging to haplotype R2 that carries small fragments of *SRKA* and *SRKC*[10]. We rechecked the bacterial artificial chromosome sequence carrying the relic *S*-locus in C24, reported in a previous study[24], and found a 434-bp putative inverted repeat sequence within the pseudo *SRKA* region (Supplementary Fig. 7a). We named the sequence *SRKIR$^{C24}$*. Using the publicly available small RNA transcriptome dataset for the C24 strain[25], we detected 3235 reads that mapped within the *SRKIR$^{C24}$* sequence in flower buds (Supplementary Fig. 7b). *SRKIR$^{C24}$* shared 99.7% sequence identity with *SRKA$^{Wei-1}$* and 84.4% sequence identity with and *ψSRKB$^{Cvi-0}$* (Supplementary Fig. 8). On the other hand, the DNA sequence identity between *SRKIR$^{C24}$* and *SRKb* was limited to 73.1%. The psRNAtarget prediction found that 329 and 20 non-redundant small RNA reads detected in *SRKIR$^{C24}$* matched perfectly to the kinase domains of *SRKA$^{Wei-1}$* and *ψSRKB$^{Cvi-0}$*, respectively (Supplementary Fig. 4). In contrast, none of these reads perfectly matched *SRKb* (Supplementary Fig. 4a), suggesting that these small RNAs may not target *SRKb*. This finding is in line with our

experimental results showing that *SRKb* can be stably expressed in the C24 background[12,19] (Fig. 1).

**A codon substituted *SRKb* is not suppressed by *SRKIR$^{Col-0}$***. The above studies suggested involvement of the sequence complementarity-based small RNA silencing pathway in the suppression of *SRKb* in Col-0. Therefore, we generated a synthetic *SRKb* sequence with synonymous substitutions in the sequence encoding the kinase domain (*synSRKb*) without altering its protein sequence and introduced this synthetic *SRKb* sequence into Col-0 (Fig. 4a). In contrast with the native *SRKb*, which shared 89.0% sequence identity with *SRKIR$^{Col-0}$*, the sequence identity between *synSRKb* and *SRKIR$^{Col-0}$* was 66.6% (Supplementary Fig. 3). The synonymous substitutions almost completely removed the predicted targetability of the small RNAs found in *SRKIR$^{Col-0}$* when examined in *synSRKb* (Supplementary Fig. 4). We hypothesized that this reduced targetability would decrease the likelihood of silencing. Lines carrying the *synSRKb* sequence exhibited full strength SI phenotypes similar to SI-C24 (Fig. 4b, c). The levels of *SRKb* mRNA were not suppressed in these lines, unlike those transformed with the native *SRKb* gene (Fig. 4d). These data suggest that the suppression of *SRKb* by *SRKIR$^{Col-0}$* requires shared sequence homology between the target and the small RNAs produced.

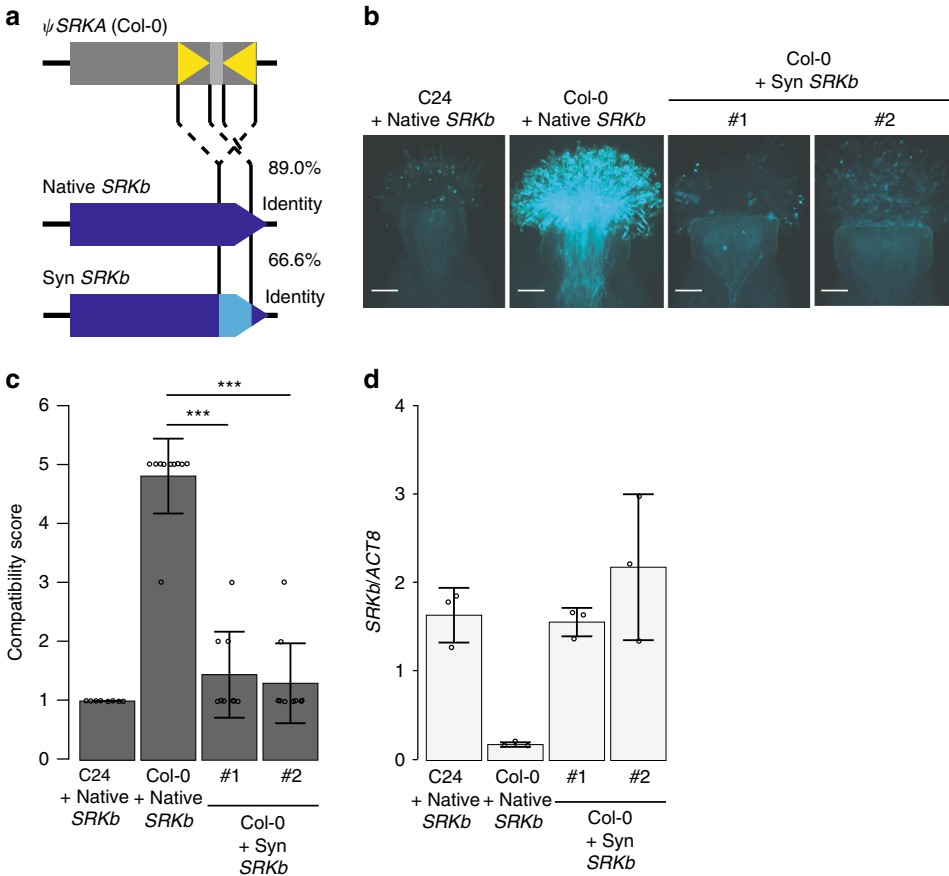

**Fig. 4. SI phenotype of Col-0 lines transformed with *SynSRKb*. a** The artificially constructed *SynSRKb* carries synonymous codon substitutions and shares 66.0% identity with *SRKIR$^{Col-0}$*, whereas the native *SRKb* shares 89.0% identity with *SRKIR$^{Col-0}$*. **b** Representative fluorescent images of aniline blue-stained pistils three hours after pollination. #1 and #2 indicate samples from two independent first generation transformants. Scale bars = 100 µm. **c** Summary of pollination tests. Statistically significant differences based on Dunnett's test are indicated by ***($p < 0.005$). The exact $p$-values were 8.55e−11 and 1.91e−11, from left to right. Error bars indicate standard deviations. Number of pistils pollinated (from left to right): $n = 8$ $n = 11$, $n = 9$, $n = 8$. **d** Relative mRNA accumulation levels of the *SRK* genes compared to *ACT8* in stigmatic tissues of the indicated lines. Error bars indicate standard deviations from independent RNA extraction trials ($n = 3$).

## Discussion

In this study, we explored the involvement of inverted repeats in relic $S$-loci of $A$. $thaliana$ in the suppression of $SRK$ mRNA accumulation. We showed that $SRKIR^{Col-0}$ can directly suppress mRNA accumulation from the introduced $SRKb$ transgene, and that this reduced transcript abundance may explain the weak reconstitution of the SI phenotype in the Col-0 strain[12]. From the finding that the promoter of $\psi SRKA$ is actively transcribed in stigma[19], it is likely that this inverted repeat is transcribed by an RNA polymerase II as is characteristic of naturally occurring hairpin RNAs[20]. It is possible that the RNA structure of $SRKIR^{Col-0}$ is processed into small RNAs by a pathway previously described in $Drosophila$[20] for naturally occurring hairpin RNAs. Further studies are necessary to understand how small RNAs are produced from $SRKIR^{Col-0}$.

Furthermore, we noticed a putative inverted repeat structure, $SRKIR^{C24}$, in the relic $S$-locus of the C24 strain. As is the case for $SRKIR^{Col-0}$, $SRKIR^{C24}$ may act directly to suppress mRNA accumulation from homologous $SRK$ sequences in a dominant fashion. The finding that $SRKIR^{Col-0}$ was able to suppress all $SRK$ alleles investigated in this study, whereas $SRKIR^{C24}$ was not able to suppress $SRKb$ may be attributable to the location of the inverted structure. $SRKIR^{Col-0}$ corresponds to a fragment encoding the kinase domain, which is relatively well conserved within the $SRK$ sequence. $SRKIR^{C24}$ was located within the variable ecto-domain, suggesting that this structure may not have the ability to suppress a broad range of diverse $SRK$. Based on these results, we propose that in silico prediction of $SRK$ alleles that can express its function when introduced into Col-0 or C24 is possible.

According to a previous study, sequence structures of the $SRKA$ regions among the A1, R0, and R1 haplotypes and those for the A2 and R2 haplotypes are largely shared[10]. Therefore, from a rough estimation, we propose that about 80.3% of $A$. $thaliana$ strains (the total of A1, A2, R0, R1, and R2 haplotypes) carry either one of these two inverted repeat types ($SRKIR^{Col-0}$ or $SRKIR^{C24}$) based on a survey of the $S$-locus sequences in 1083 strains[10]. Several studies suggest that the initial step in the evolution of self-compatibility in the A haplotype of $A$. $thaliana$ was a loss-of-function of the $SP11/SCR$ gene because functional forms of this gene are absent from all strains investigated[8,10,11].

Whether the dominant $SRK$ suppressors played a role in the evolution of SC in $A$. $thaliana$ is an intriguing question. In the genus $Brassica$, $S$-locus-linked small non-coding RNA genes in the dominant alleles silence mRNA expression of the recessive $SP11/SCR$[26,27]. Similarly, several $SP11/SCR$ dominance classes are predicted in species such as $A$. $lyrata$[22,28] and $A$. $halleri$[29]. Further study of the $A$. $halleri$ system found that there are at least 17 small RNA-producing loci in the region that contribute to the establishment of dominance relationships[30]. It has been argued that gene-disrupting mutations of $SP11/SCR$ or $SRK$ in dominant $S$-haplogroups are predicted to be more likely spread than recessive mutations when SC is favored[8,31]. It is possible that in certain climatic situations such as the glacial periods when pollen availability is limited[11,32], such a dominant $SRK$ suppressor mutation may spread throughout the population faster than a recessive mutation. More theoretical studies on the evolution of SC by incorporating the allelic dominance effect would be worthwhile using existing frameworks[32,33].

The fact that $SRKIR^{Col-0}$ and $SRKIR^{C24}$ independently evolved twice in the evolution of $A$. $thaliana$ is reminiscent of the classical example of the spread of a dominant phenotype: the melanization of the peppered moth ($Biston$ $betularia$) during the Industrial Revolution[34]. The melanic phenotype is dominant over other color types and also independently evolved twice in this species[35]. Our results support the model that inverted repeats are an easy way to establish an allelic dominance relationship. It is possible that inverted repeat structures may explain other dominant-recessive genetic systems.

## Methods

**Plant materials.** All plant materials were grown in mixed soil in a growth chamber under controlled conditions (light intensity, 120–150 μmol m$^{-2}$ s$^{-1}$; 14-h light/10-h dark cycle; 22 ± 2 °C). The T-DNA insertion line of $SRKIR^{Col-0}$ (SALK_137645C) was obtained from the Arabidopsis Biological Resource Center. $A$. $lyrata$ SaSb seeds were gifts from Dr. Yoshinobu Takada, Tohoku University.

**Transgenic experiments.** The $SRKb$-$SCRb$/pBI121 construct was produced by the below procedure. The $S_b$-$SP11/SCR$ and $S_b$-$SRK$ genes were derived from the $S_b$-haplotype of self-incompatible $A$. $lyrata$[19]. $S_b$-$SP11/SCR$ was fused to the promoter region of $B$. $rapa$ $S_9$-$SP11/SCR$[36]. Construct for expressing $S_b$-$SP11/SCR$ ($SCRb$/pBI121) was generated in the previous study[19]. For $S_b$-$SRK$ expression in papilla, the $\Psi SRKA$ upstream region was cloned from Col-0 and used as the promoter sequence to enable papilla–cell specific expression. $\Psi SRKA$ upstream region was amplified using the primers listed in Supplementary Table 1. The PCR fragment was cloned into pGEM T-Easy vector (Promega) using the manufacturer's protocol ($\Psi SRKA$/pGEM). The $NOS$ terminator was amplified from pBI121 using the primers listed Supplementary Table 1, and then was cloned into pGEM T-Easy vector to obtain $NOS$/pGEM. Both $\Psi SRKA$/pGEM and $NOS$/pGEM plasmids were digested with $Kpn$I (Takara) and $Sac$II (Takara), and the $NOS$ terminator fragment was inserted into $\Psi SRKA$/pGEM to obtain the $\Psi SRKA$-$NOS$/pGEM plasmid. The $SRKb$ coding fragment was amplified with primers listed in Supplementary Table 1, and then cloned into the $Kpn$I site of the $\Psi SRKA$-$NOS$/pGEM plasmid to obtain $\Psi SRKA$-$SRKb$-$NOS$/pGEM. The $\Psi SRKA$-$SRKb$-$NOS$ cassette was digested out with $Eco$RI (Takara), and then cloned into the $Eco$RI site of the $SCRb$/pBI121 plasmid to obtain $SRKb$-$SCRb$/pBI121.

The synSRKb sequence (Supplementary Fig. 9) was chemically synthesized by Medical & Biological Laboratories. The $SRKb$-$SCRb$/ pBI121construct was digested with $Kpn$I to cut out the $SRKb$ moiety and was treated with calf intestine alkaline phosphatase (Takara). The synSRKb fragment PCR amplified by primers listed in Supplementary Table 1 to add the $Kpn$I restriction site. Amplified fragment was digested by $Kpn$I and then introduced into the vector.

A 5,353-bp genomic fragment containing the full $SRKIR^{Col-0}$ sequence or the fragment containing the partial $SRKIR^{Col-0}$ sequence was amplified by PCR using Col-0 DNA as the template with the primers listed in Supplementary Table 1. Fragments were introduced into pCambia1300 digested with $Hin$dIII (Takara) and $Sac$I (Takara) using the In-Fusion HD Cloning Kit (Takara-Bio, Kusatsu, Japan). These constructs were introduced into the SI-C24 line. Five independent $T_1$ plants introduced with partial $SRKIR^{Col-0}$ were obtained. Since they all expressed strong SI comparable to SI-C24, we analyzed one out of them for $SRKb$ transcript accumulation.

All transgenic plants were generated using the $Agrobacterium$ infiltration procedure, as previously reported[37]. Plants transfected with pBI121 binary vectors were selected for kanamycin resistance and pCambia1300 were selected for hygromycin resistance.

**Genetic analysis of the T-DNA insertion line.** The Col-0 + $SRKb$-$SCRb$ line was crossed with the homozygous T-DNA insertion line SALK_137645C to obtain an $F_1$ population. Individuals from the segregating $F_2$ generation were genotyped for the presence of $SRKb$, and then genotyped for the presence of T-DNA insertions. Primers used for genotyping are listed in Supplementary Table 1.

**Pollination experiments.** Flowers were emasculated before anthesis. Pistils of flowers at stage 14 were harvested, transferred to 1% (w/v) agar plates, pollinated with pollen from stage 13 flowers, and then incubated for three hours under controlled conditions (22 ± 2 °C, humidity 50 ± 5%). For Supplementary Fig. 1, pistils from stage 13 flowers were also used in pollination tests. The pollinated pistils were fixed overnight at room temperature in a freshly prepared ethanol/ acetate 3:1 (v/v) solution. After replacing the solution with 1 M sodium hydroxide, the pistils were incubated at 60 °C for 30 min ($A$. $thaliana$) or 60 min ($A$. $lyrata$). The solution was then replaced with the aniline blue staining solution (2% (w/v) tripotassium phosphate, 0.01% (w/v) aniline blue) and incubated at room temperature for 3 h. To facilitate phenotyping, we defined arbitrary compatibility scores based on the number of pollen tubes in the styles: 1, no tubes observed; 2, 1–10 tubes; 3, 11–30 tubes; 4, 31–60 tubes; 5, ≥60 tubes.

**Gene expression studies.** Total RNA was extracted from stigmas at stage 13 with the RNeasy Plant Mini Kit (Qiagen, Hilden, Germany). cDNA was synthesized from the RNA using the SuperScript IV Reverse Transcriptase Kit (Thermo Fischer Scientific, Massachusetts, USA). The real-time PCR reactions were performed with the QuantiTect SYBRGreen PCR Kit (Qiagen) using the LightCycler 96 system (Roche, Basel, Switzerland). Relative quantification was calculated relative to the endogenous $ACTIN8$ ($ACT8$: AT1G49240) gene, using the Relative quantification application mode of the LightCycler 96 system. The same set of

primers was used to quantify the *ACT8* ortholog from *A. lyrata* (genesh2_kg. 1__4019__AT1G49240.1). Primers used for the quantitative reverse transcription-PCR analyses are listed in Supplementary Table 1.

**Small RNA transcriptome analysis**. Small RNAs were extracted from Col-0 stigmas at stage 13 using the mirVana™ miRNA Isolation Kit (Ambion). Library preparation and Illumina Hiseq single-end sequencing were conducted by Hokkaido System Science Co., Ltd. Small RNA transcriptome data for the C24 flower buds (under run IDs SRR4026067 and SRR4026068) were downloaded from the Sequence Read Archive at the National Center for Biotechnology Information [NCBI]. After adaptor trimming with the cutadapt version 1.1 software[38] using parameters -O 5/-e 10/-g GTTCAGAGTTCTACAGTCCGACGATC/-a TGGAATTCTCGGGTGCCAAGG and rest set as defaults. The reads were mapped to the *Arabidopsis* genome (version TAIR10) or within the bacterial artificial chromosome sequence of the *S* locus in C24 (NCBI ID: EF182720) with Bowtie version 1.0.0[39] using parameters -v 2/-a/--best/--strata/-m 20 and rest set as defaults. Mapped reads were extracted using the functions implemented in the SAMtools suite[40] with default parameters. The results were visualized with the Integrated Genome Viewer 2.3.93[41].

**Small RNA target prediction**. Small RNA targetability predictions within the *SRK* sequences (*SRKb*; AB052756, *SRKA*$^{Wei-1}$; GU723787, *ψSRKB*$^{Cvi-0}$; AY772644) were evaluated using the psRNAtarget program[21] with default parameters and an expectation score of 3.0 as a cut-off. An expectation score of 0 in this program indicates a perfect match; prediction scores ranging from 0.5 to 3.0 indicate potential targets with mismatches. From the small RNAs mapped to *SRKIR*$^{Col-0}$ or *SRKIR*$^{C24}$, those sized from 19 to 24 nucleotides (typical sizes for those functioning in gene silencing) were used for the analysis. Non-redundant reads were selected using the Fastx_Collapser program[42].

**Statistical analyses and exact p values**. All of the statistical analyses and bar plot visualizations were carried out using R[43]. Dunnett's test was performed using the glht function from the multcomp package[44]. For all bar plots, the values indicate means and the error bars indicate standard deviations. The number of replicates and the exact *p*-values are defined in the figure captions.

**Reporting summary**. Further information on research design is available in the Nature Research Reporting Summary linked to this article.

## Data availability

Raw sequence data files were deposited at the National Center for Biotechnology Information Sequence Read Archive (NCBI-SRA) under BioProject ID PRJNA561260. The data underlying Figs. 1a, 1b, 1c, 2b, 2c, 2d, 3a, 3b, 3c, 3d, 3e, 4b, 4c and 4d as well as Supplementary Figs. 1, 5b, 5c and 5d are provided as a Source Data file. All other data are available in the manuscript or in the supplementary materials.

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

## Acknowledgements

We thank M. Okamura, M. Nara, T. Manabe, and Y. Yamamoto for their technical assistance. This work was supported in part by Grants-in-Aid for Scientific Research on Innovative Areas (23113002, 16H06467, 16H06464 to S.T.; 16H01467, 18H04776 to S.F.), Grants-in-Aid for Scientific Research (21248014, 25252021, 16H06380 to S.T.; 18H02456 to S.F.), a Grant-in-Aid for Challenging Exploratory Research (15K14626 to S.F.) from the Ministry of Education, Culture, Sports, Science and Technology of Japan (MEXT) and the Japan Science and Technology Agency (JST) PRESTO program (JPMJPR16Q8) to S.F.

## Author contributions

S.T. conceived the study. S.F and S.T. wrote the paper. S.F. and H.S.-A. conducted the majority of the experiments and analyzed the data. M.K., T.K., M.I. contributed to transgenic experiments.

## Competing interests

The authors declare no competing interests.
