## [Peer Review File · Nature Communications]

Reviewers' comments:

Reviewer #1 (Remarks to the Author):

The manuscript by Fujii et al. details an interesting piece of work challenging the popular view that loss-of-function mutations are recessive over functional alleles. The authors propose a small RNA-based mechanism by which two different non-functional alleles of the self-incompatibility (SI) gene SRK in *Arabidopsis thaliana* rather behave in a dominant manner. The starting point is the intriguing observation that *A. thaliana* accessions vary in their capacity to express self-incompatibility (SI) following genetic transformation with the SI genes from the closely related outcrossing species *A. lyrata*: some accessions, like C24 express relatively strong SI, while others like Col-0 remain selfers. The causes of this variation have been previously debated in the literature, but have remained largely elusive. Here, the authors hypothesize that the difference between Col-0 and C24 is due to the presence of hairpins formed by independent inversions of parts of the SRK gene (a 502bp inverted repeat in the kinase domain in Col-0 vs a 434bp inverted repeat in the extracellular domain in C24) that have the capacity to recognize and suppress some (but not all) SRK mRNAs even when the hairpin is present as a single copy.

While I found the hypothesis interesting and the results clearly worth reporting, the manuscript suffers from a number of shortcomings that should be addressed. First, the methods are very incomplete at some places. Second, the presentation of the broad context of the study and of the implications of the results is also incomplete and important pieces of information on the study system are missing. As a result, the manuscript is currently frustrating on both accounts: on the insight gained on self-incompatibility itself and on the description of the proposed mechanism of gene regulation. I am listing below several Major (M) and minor (m) comments to improve the presentation. I think that most of them can be addressed quite readily and I have attempted to formulate them in details, but there are many of them. I would be happy to review a revised version taking them into account.

- [M] There is an important omission in how the context is presented, since a number of other molecular mechanisms have been proposed to explain so-called « dominant-negative » mutations (reviewed some time ago by Veita 2007, <https://www.ncbi.nlm.nih.gov/pmc/articles/PMC2217636/>). The results of the present study should be placed in that broader context, not just that of Fisher's genetic elements.

- [M] Important details about the description of the hairpins are missing. In particular:

o The claim that the hairpin in Col-0 produces sRNAs is premature in the absence of a formal test of whether they might have been produced by a different locus in the genome. Testing whether those sRNAs map uniquely to the hairpin is required. This is a concern since the hairpin in Col-0 is formed in the kinase domain of SRK, and kinase domains are ubiquitous across the genome. I would suggest to remove any small RNAs that are not matching perfectly and uniquely to the hairpin when aligned across the whole genome.

o Does the hairpin in Col-0 cover just a single exon of the kinase domain, or does it span over intronic sequences as well? If so, does the primary transcript (before cleavage into small RNAs) also include those intronic regions even though they are not expected to have any target on the (spliced) mRNA?

o In terms of evolutionary steps involved, how would the hairpin acquire transcriptional activity? Since it was formed from the kinase domain in Col-0, it is not expected to contain any transcription start site to begin with and must have acquired it subsequently. This issue should at least be discussed somewhere in the manuscript.

- [M] The assertion that the silencing mechanism is based on post-transcriptional gene silencing (PTGS) is based on the size distribution only, which is very weak evidence in itself.

o I would suggest that rather than making this assertion the authors acknowledge that in the absence of further experiments (which would be relatively straightforward in Col-0) the exact mechanism cannot be determined.

- o Another possibility could be to look for any bias in the identity of the 5' nucleotide, which bears some correlation with the ARGONAUTE protein into which they will be sorted (<https://www.ncbi.nlm.nih.gov/pmc/articles/PMC2981139/>).
- o the peak of 20nt molecules in the size distribution of small RNAs is unexpected. For classical microRNAs, most molecules will rather be 21nt long. It is essential that the authors show the size distribution of small RNAs across the whole genome, not just the hairpin to verify that the expected peaks of 21 and 24nt molecules are indeed observed and there is no problem with the sequencing experiment itself.
- o Sequence similarity between the hairpin and the putative target region is measured using a vague and arbitrary overall comparison. There are a number of bioinformatic tools available to predict the targets of microRNA (<https://bmcmgenomics.biomedcentral.com/articles/10.1186/1471-2164-15-348>). The authors should either use them or explain why they decided otherwise.

- [M] The manuscript is very unclear and even misleading about the structure of polymorphism of the S-locus :

- o First, it should be made clear in line 95 that the 12 haplotypes described in Tsuchimatsu et al. (2017) actually form three main divergent haplogroups, and the 12 haplotypes are derived from different rearrangements of these three initial haplotypes. Of the accessions included in the present manuscript, Col-0, Uk-3 and Wei-1 belong to haplogroup A, Cvi-0 belong to haplogroup B, and C24 belongs to haplotype R3 formed by the recombination between a A and a C haplotype followed by deletion of the main part of the A haplotype (see Fig S2 in Tsuchimatsu et al. 2017). The small remaining part of SRK sequence in C24 therefore derives from the C haplogroup. This information should be reported.
- o Second, it should also be made clear that the three haplogroups (A, B and C) derive from a much larger set of diverged allelic lines in the closely related outcrossers, among which the SRKb sequence introduced by transformation is yet another divergent lineage.
- o The fact that the Col-0 hairpin seems to be able to suppress all SRK alleles introduced while the C24 hairpin suppresses all of them but SRKb is an interesting observation whose evolutionary significance could be given more importance in the manuscript. I see two non-exclusive explanations. First, it could be due to the location of the hairpin (in the more conserved kinase domain for Col-0 but in the more variable extracellular domain for C24). Second, it could be due to the initial level of dominance of the S-allele onto which the hairpin arose. Since the Col-0 S-allele is relatively lower in the dominance hierarchy (haplogroup A) than the C24 S-allele (haplogroup C for the SRK chunk), at least as can be predicted based on the phylogenetic position of their *A. halleri* and *A. lyrata* orthologs, selection for silencers of SRK in heterozygous plants could be stronger on the more recessive haplotypes (A) than on the more dominant haplotypes (C) in which the small RNA-based mechanism controlling the dominant pollen phenotype would already be inactivating expression of the SCR allele in the other chromosome. A more complete crossing design to compare the silencing capacity of the hairpins of Col-0 and C24 towards a broader range of SRKs would be a quite interesting follow-up experiment. I don't think the authors should be required to perform that experiment in the present study, but it might be worth expanding on the topic a little bit in the text. More generally, the evolutionary scenario for how dominance could have facilitated the loss of SI should be laid out more explicitly in the discussion.

- [m] important details about the transgenic lines are missing (line 81). In particular some details about the SRKb-SCRb transformation should be given : are they introduced as a single plasmid or by separate transformation experiments ? How were the cloned fragments determined ?

- [m] whether these results are consistent with Liu et al. (2007) on the role of PUB8 variation should be made explicit.

- [m] The size of the genomic fragment introduced in C24 to validate the role of the Col-0 hairpin is quite large (5353bp) and thus contains more than just the hairpin. This raises the possibility that other nearby elements in the fragment may be causing the phenotype. This limitation should be acknowledged and the reason for using such a large fragment should be given. Also, the

primers used for cloning and the detailed transformation protocol should be reported.

- [M] The protocol to evaluate transcript abundance is not even mentioned and should be fully detailed. Where along the gene were the qPCR primers designed and how were they tested and optimized? How many biological/technical replicates? A specific concern is whether the primers are specific across the different SRK alleles. How have the authors validated their specificity?

- [M] Along the same line, the statistical method to compare transcript abundance in Fig. 3d and 3e is not mentioned. As I understand, the authors are just doing visual comparison, but the fact that the wildtype ecotypes are homozygotes while the hybrids are heterozygotes renders the comparison far from obvious (a ratio of 1/2 should be expected even in the absence of any suppression, and should be used as the null hypothesis). It is essential that the statistical method is improved. Also, the label of the Y axis is unclear: does that correspond to ratios of Ct values? In the legend, the word "expression" is not correct: what is measured is transcript levels. In Fig 3b I think the X axis is mislabelled. The second column should read C24/Sb and the third Col-0/Sb.

- [m] The role of the C24 hairpin is not established as firmly as that the Col-0 hairpin, as not functional validation by genetic transformation was performed. Also, whether this other hairpin also produces small RNA was not tested. This should be straightforward to determine since flower buds sRNA data are available from the literature (see e.g. <https://www.ncbi.nlm.nih.gov/geo/query/acc.cgi?acc=GSE85551>, although the exact stage might not be appropriate). At the very least this limitation should be clearly acknowledged.

- [M] The last paragraph mentions "the evolution of a broader epigenetic regulatory system" and links this to "the formation of micro RNA genes". This wording is inappropriate on several grounds. One, at present there is no reason to call the suppression phenomenon "epigenetic", as this is mostly presented as classical genetic regulation (PTGS) in the manuscript. Second, beside their size (20nt, which is unusual by the way), the link with microRNA genes is not clear. Canonical microRNA genes in plants are typically much shorter hairpin structures producing a largely unique (mature) 21nt molecule, which is very different from the pattern observed here. I would suggest to simply remove this sentence.

- [m] The layouts of Fig 1a and Fig S1 are different. It would be a lot easier to follow if they were identical.

- [M] In Figure 1a, an important control is missing, to show that pollen from Col-0 SRKb-SCRb is indeed fertile and able to germinate on stage 13 stigmas.

- [M] The method for the pollination experiments "other than Fig 1" is described in details, but nothing is reported for Fig. 1. Please clarify.

- [m] Line 87-88, the stage at which the pollination experiments were performed should be given in the text, not just the figure legend.

- [m] A consequence of the model proposed is that based on sequence information alone, one should be able to predict whether a given SRK allele can be successfully introduced into a given *A. thaliana* accession by transformation. This conclusion could perhaps be added to the discussion.

Review by Vincent Castric

Reviewer #2 (Remarks to the Author):

This manuscript provides evidence suggesting that suppression of function of self-incompatibility in

Arabidopsis thaliana can be driven by post-transcriptional gene silencing mediated through small RNAs. The manuscript is well-written, the results are interesting and the experiments conducted appropriately but I wasn't convinced that there is a substantive enough gain in knowledge to warrant publication in a high impact general journal. The short format also means that the story is not really told with sufficient background for a general audience to understand the context or the novelty and the methods are too brief to fully explain what was done.

I have made specific suggestions on an annotated version of the manuscript but my major concern is the presentation of the context and novelty, not only for a general audience but also for experts.

1) The presentation of background literature is biased to a few studies from a few laboratory groups, rather than presenting the broader body of literature suggesting possible mechanisms for loss of function of self-incompatibility in *A. thaliana* and its close relatives (including the extensive body of work on Brassicas).

2) There is also not enough context set about the background within *A. thaliana* in the introduction; more details about the variation in SI alleles is presented in the methods and results but it would help the reader and be more logical to provide all of the background to motivate the study in the introduction.

3) The reader is also not told until the discussion that the inversions that are the focus of the paper had already been described and their frequency across accessions of *A. thaliana* quantified in a previous study. This compromises the novelty of the paper as currently presented. However, with refocusing, adding value to the previous study by the experimental testing in this study could broaden the impact of the paper.

4) The paper generally lacks a logical flow of information. Various hypotheses are suggested in the methods, without explaining the specific logic that motivated them; it would be clearer to motivate the hypotheses in the introduction and then use the methods and results to test them. I have noted specific examples on the annotated version of the manuscript. It is quite confusing whether the hypotheses emerged from this study or were motivated by previous results.

5) There is effectively no description of the statistics and there is a lack of quantitative details throughout the results (e.g. "high" sequence similarity).

6) I do appreciate that the short format means that describing some methods in the results is necessary but the results are dominated by methods and the methods are too brief to allow repeatability.

Reviewer #3 (Remarks to the Author):

This study by Fujii et al., 'Parallel evolution of dominant pistil-side self-incompatibility suppressors in *Arabidopsis*' identifies inverted repeat sequences in different regions of relic or pseudo SRK genes in Col-0 and C-24 respectively, that are responsible for silencing SRK genes through PTGS. They propose that these evolved in a parallel fashion and in populations where SC is favoured these mutations may have spread faster than a recessive mutation in SRK.

The work primarily focused on the discrepancy of why SI response in Col-0 was transient while in C24 it was more stable. The authors show that SRKb is efficiently targeted by the 502 bp inverted repeat corresponding to the kinase domain of SRKb. They also find a second mutation in C24, a 434-bp putative inverted repeat sequence within the pseudo SRK region of C24 (SRKIRc24) and this has low similarity with SRKb thereby, not affecting SRKb expression.

Although the idea of inverted repeats leading to silencing is not novel, this is the first report explaining why Col-0 plants show transient SI when SRKb/SCRb are expressed. While overexpression and expressing a synthetic version are quite promising, the ultimate proof would be to eliminate this 502 bp repeat sequence to show that a strong SI response can be conferred by expression of the SRKb/SCRb combo. The authors should genome edit Col-0 to eliminate the 502 bp region to show that SRKb can be more stable and manifest a strong SI response in the gene-edited (502 bp removed) background.

Other minor points:

1. Supplementary Fig. 1, The authors say they generated these lines (Col-0 + SRKb-SCRb). Are these pre-existing lines or newly created transgenic lines?
2. Figure 2, What promoter was used when introducing SRKIRcol-0 in to C24 background? Were any other SRK-related or S-domain kinases suppressed? Please show suppression of SRKb in these transgenic lines.
3. Figure 3, while SCRb is dominant over SCRa, the SRKIRcol-0 inverted repeat should target both SRKa and SRKb in these lines. What is the expression level of SRKa in these lines (Col-0 and C-24 expressing Sa) in spite of being compatible with pollen from *A. lyrata* SaSb. We would expect suppression of SRKa in Col-0/Sa line.
4. Figure 4, what promoter was used to express the synthetic SRKb gene in Col-0 background? As a control native gene under the same promoter should be expressed in Col-0 background to examine if it can be suppressed by SRKIRcol-0 and a compromise in SI should be observed.
5. The authors should discuss the parallel evolution, is it parallel with the inverted repeat identified in the pollen SCR fragments or between the inverted repeats of SRK in Col-0 and C-24

Reviewers' comments:

Reviewer #1 (Remarks to the Author):

The manuscript by Fujii et al. details a interesting piece of work challenging the popular view that loss-of-function mutations are recessive over functional alleles. The authors propose a small RNA-based mechanism by which two different non-functional alleles of the self-incompatibility (SI) gene SRK in *Arabidopsis thaliana* rather behave in a dominant manner. The starting point is the intriguing observation that *A. thaliana* accessions vary in their capacity to express self-incompatibility (SI) following genetic transformation with the SI genes from the closely related outcrossing species *A. lyrata* : some accessions, like C24 express relatively strong SI, while others like Col-0 remain selfers. The causes of this variation have been previously debated in the literature, but have remained largely elusive. Here, the authors hypothesize that the difference between Col-0 and C24 is due to the presence of hairpins formed by independent inversions of parts of the SRK gene (a 502bp inverted repeat in the kinase domain in Col-0 vs a 434bp inverted repeat in the extracellular domain in C24) that have the capacity to recognize and suppress some (but not all) SRK mRNAs even when the hairpin is present as a single copy.

While I found the hypothesis interesting and the results clearly worth reporting, the manuscript suffers from a number of shortcomings that should be addressed. First, the methods are very incomplete at some places. Second, the presentation of the broad context of the study and of the implications of the results is also incomplete and important pieces of information on the study system are missing. As a result, the manuscript is currently frustrating on both accounts: on the insight gained on self-incompatibility itself and on the description of the proposed mechanism of gene regulation. I am listing below several Major (M) and minor (m) comments to improve the presentation. I think that most of them can be addressed quite readily and I have attempted to formulate them in details, but there are many of them. I would be happy to review a revised version taking them into account.

- [M] There is an important omission in how the context is presented, since a number of other molecular mechanisms have been proposed to explain so-called « dominant-negative » mutations (reviewed some time ago by Veita

2007, <https://www.ncbi.nlm.nih.gov/pmc/articles/PMC2217636/>). The results of the present study should be placed in that broader context, not just that of Fisher's genetic elements.

Response: We thank the reviewer for suggestion. We cite the paper in this version to improve the introduction in this paper.

- [M] Important details about the description of the hairpins are missing. In particular:

o The claim that the hairpin in Col-0 produces sRNAs is premature in the absence of a formal test of whether they might have been produced by a different locus in the genome. Testing whether those sRNAs map uniquely to the hairpin is required. This is a concern since the hairpin in Col-0 is formed in the kinase domain of SRK, and kinase domains are ubiquitous across the genome. I would suggest to remove any small RNAs that are not matching perfectly and uniquely to the hairpin when aligned across the whole genome.

Response: We understand this concern by the reviewer here. Actually, 99.5% of the reads (25563 out of 25671) were uniquely mapped to this inverted repeat, meaning it is highly likely that these small RNAs are indeed produced from the hairpin. We added this statement to the results section describing the small RNA transcriptome in the current version of the manuscript.

o does the hairpin in Col-0 cover just a single exon of the kinase domain, or does it span over intronic sequences as well ? If so, does the primary transcript (before cleavage into small RNAs) also include those intronic regions even though they are not expected to have any target on the (spliced) mRNA ?

Response: The inverted repeat covered exons 6 and 7. We found small RNA reads also map to the intronic regions, albeit with less abundance compared to the exon regions. We added Supplementary Figure 5 that visualize this.

o in terms of evolutionary steps involved, how would the hairpin acquire transcriptional activity ? Since it was formed from the kinase domain in Col-0, it is not expected to contain

any transcription start site to begin with and must have acquired it subsequently. This issue should at least be discussed somewhere in the manuscript.

Response: We supplied detailed explanation of the possible transcriptional status of this gene locus. The promoter of AT4G21370, the pseudo *SRKA* gene in Col-0, is active in stigmas. In a past study, we fused this promoter to the *SRKb* cDNA cloned from *A. lyrata*, and successfully reconstituted SI in *A. thaliana* by introducing this construct (Iwano et al 2015). This is also supported by the TRAVA database (<http://travadb.org/browse/DeSeq/At4g21370/RawNorm/AvNorm/Color=RCount/>). Thus, we believe that this hairpin is transcribed by polymerase II, similar to microRNA genes. We speculate that AT4g21370 is transcribed by pol II in the first paragraph of the discussion.

- [M] The assertion that the silencing mechanism is based on post-transcriptional gene silencing (PTGS) is based on the size distribution only, which is very weak evidence in itself. o I would suggest that rather than making this assertion the authors acknowledge that in the absence of further experiments (which would be relatively straightforward in Col-0) the exact mechanism cannot be determined.

Response: We understood the concern here by the reviewer that a support for possible silencing mechanism is missing. We deleted the relevant sentence and instead discussed that further experiments are necessary to understand the silencing mechanism.

o Another possibility could be to look for any bias in the identity of the 5' nucleotide, which bears some correlation with the ARGONAUTE protein into which they will be sorted (<https://www.ncbi.nlm.nih.gov/pmc/articles/PMC2981139/>).

Response: As suggested, we looked into our data and sought for a bias in the presence of the 5' nucleotides. Although we found that small RNAs starts with 'A' were slightly abundant (33%) compared to other nucleotides in the *SRKA* hairpin, this was not significantly different with the entire transcriptome ('A' accounted for 37%). At present we consider it is difficult to specify the AGO species involved in the silencing.

o the peak of 20nt molecules in the size distribution of small RNAs is unexpected. For classical microRNAs, most molecules will rather be 21nt long. It is essential that the authors show the size distribution of small RNAs across the whole genome, not just the hairpin to verify that the expected peaks of 21 and 24nt molecules are indeed observed and there is no problem with the sequencing experiment itself.

Response: We investigated the read size distribution of our raw data. This new data is in Supplementary Figure 2b. After filtering out the reads that map to tRNA, we found that 24-nt small RNA was the most abundant in our transcriptome (the whole genome sample in Supplementary Figure 2). Since small RNA size populations are known to greatly vary in between different tissues, genotypes and growth conditions, the profile we found is within the range of variation found in numerous small RNA transcriptomes studies in Arabidopsis. We thus believe that our sequence experiment is valid.

o Sequence similarity between the hairpin and the putative target region is measured using a vague and arbitrary overall comparison. There are a number of bioinformatic tools available to predict the targets of microRNA

(<https://bmcgenomics.biomedcentral.com/articles/10.1186/1471-2164-15-348>). The authors should either use them or explain why they decided otherwise.

Response: Taking account this advice, we used the program psRNAtarget to predict targetability of *SRKb*, *SRK^{Cvi-0}* and *SRK^{Wei-1}* by the small RNAs found from the inverted repeat regions. The result clearly showed that small RNA that may be produced from the inverted repeats of Col-0 can, but those produced from C24 cannot target *SRKb*. In the mean time, small RNAs produced from both can target *SRK^{Cvi-0}* and *SRK^{Wei-1}*, as expected from the qRT-PCR data. These series of data are newly added to this version of the manuscript.

- [M] The manuscript is very unclear and even misleading about the structure of polymorphism of the S-locus :

o First, it should be made clear in line 95 that the 12 haplotypes described in Tsuchimatsu et al. (2017) actually form three main divergent haplogroups, and the 12 haplotypes are derived from different rearrangements of these three initial haplotypes. Of the accessions included in the present manuscript, Col-0, Uk-3 and Wei-1 belong to haplogroup A, Cvi-0 belong to haplogroup B ,and C24 belongs to haplotype R3 formed by the recombination between a A and

a C haplotype followed by deletion of the main part of the A haplotype (see Fig S2 in Tsuchimatsu et al. 2017). The small remaining part of SRK sequence in C24 therefore derives from the C haplogroup. This information should be reported.

o Second, it should also be made clear that the three haplogroups (A, B and C) derive from a much larger set of diverged allelic lines in the closely related outcrossers, among which the SRKb sequence introduced by transformation is yet another divergent lineage.

Response: We agree that such background introductions were lacking in the previous version of the manuscript, as also pointed out by reviewer 2. This information (Wei-1/Uk-3 belong to haplogroup A, Cvi-0 to B) is now reported in the introduction of the current version.

However, there is one thing we wish to confirm with reviewer 1, who presumably is one of the authors of the relevant paper Tsuchimatsu et al 2017. According to Fig. 6 from the Tsuchimatsu paper, C24 belongs to haplotype R2 and not R3. The R2 haplotype includes small fragment of SRKA (according to Fig. S2 from the same paper). This is agreed by the BAC sequence of C24 (Sherman-Broyles et al 2007, EF182720). We found that the SRKA moiety (corresponds to ecto-domain of SRK) forms an inverted repeat ($SRKIR^{C24}$), and small RNAs were mapped to there (Supplementary Fig. 7). The current version of the manuscript introduces these in the results.

o The fact that the Col-0 hairpin seems to be able to suppress all SRK alleles introduced while the C24 hairpin suppresses all of them but SRKb is an interesting observation whose evolutionary significance could be given more importance in the manuscript. I see two non-exclusive explanations. First, it could be due to the location of the hairpin (in the more conserved kinase domain for Col-0 but in the more variable extracellular domain for C24). Second, it could be due to the initial level of dominance of the S-allele onto which the hairpin arose. Since the Col-0 S-allele is relatively lower in the dominance hierarchy (haplogroup A) than the C24 S-allele (haplogroup C for the SRK chunk), at least as can be predicted based on the phylogenetic position of their *A. halleri* and *A. lyrata* orthologs, selection for silencers of SRK in heterozygous plants could be stronger on the more recessive haplotypes (A) than on the more dominant haplotypes (C) in which the small RNA-based mechanism controlling the dominant pollen phenotype would already be inactivating expression of the SCR allele in the other chromosome. A more complete crossing design to compare the silencing capacity of the hairpins of Col-0 and C24 towards a broader range of SRKs would be a quite interesting follow-up experiment. I don't think the authors should be required to perform that experiment in the present study, but it might be worth expanding on the topic a little bit in the text. More generally, the evolutionary scenario for how dominance could have facilitated the loss of SI should be laid out more explicitly in the discussion.

Response: As suggested here, we added the possible explanation to the fact that Col-0 hairpin broadly suppress SRK diversities, while that of the C24 type did not suppress SRKb, to the discussion. We believe that this is simply due to the fact that the Col-0 type corresponds to the more conserved kinase domain, and the C24 type is located on the ecto-domain. This was predicted so by the psRNAtarget program suggested by this reviewer (Supplementary Fig. 4).

The alternative explanation by the reviewer here: "selection for silencers of SRK in heterozygous plants could be stronger on the more recessive haplotypes (A) than on the more dominant haplotypes (C) in which the small RNA-based mechanism controlling the dominant pollen phenotype would already be inactivating expression of the SCR allele in the other chromosome." is a very attractive interpretation and may give a biological significance to what we observed in this study. However, we felt it is a bit of an over-discussion to include such hypothesis in our discussion here, given that *SP11/SCR* dominance hierarchy in *A. thaliana* is not understood and is based on the speculation from the sequence conservation with the *A. halleri* system. We instead included previous citations in the introductions that argued, gene-disrupting mutations of SI in dominant S-haplogroups are predicted to be more likely spread than recessive mutation (by Kentaro Shimizu group).

- [m] important details about the transgenic lines are missing (line 81). In particular some details about the SRKb-SCRb transformation should be given : are they introduced as a single plasmid or by separate transformation experiments ? How were the cloned fragments determined ?

Response: As suggested, procedure for obtaining the genomic constructs including *SRKb-SCRb* are now described in the methods. In brief, we inserted the *SRKb* expressing cassette to the construct for expressing *S_b-SP11/SCR (SCRb/pBI121)* generated in our previous study (Iwano et al. 2015), to obtain *SRKb-SCRb/pBI121* that expresses both components.

- [m] whether these results are consistent with Liu et al. (2007) on the role of PUB8 variation should be made explicit.

Response: This is made clear in the relevant section. Our result disagree with Liu et al. (2007).

Liu et al. (2007) analyzed F₂ of Col-0 x C24 and reported that gene locus harboring *PUB8* and *SRKIR^{Col-0}* is responsible for the weak SI phenotype in Col-0. They reported that the Col-0 allele is recessive for conferring pseudo self-incompatibility. They introduced the C24 form of *PUB8* to Col-0 and found that some transgenic lines exhibits reduced pollen tube growth by self-pollinations.

In this work, we found that Col-0 carries a dominant endogenous factor, not present in C24, that suppresses the expression of *SRK* (Figure 1). This recessive or dominant disagreement is stated in the first section of the results. We also showed that introduction of *SRKIR^{Col-0}* was sufficient to breakdown SI in the C24 background (Figure 2).

- [m] The size of the genomic fragment introduced in C24 to validate the role of the Col-0 hairpin is quite large (5353bp) and thus contains more than just the hairpin. This raises the possibility that other nearby elements in the fragment may be causing the phenotype. This limitation should be acknowledged and the reason for using such a large fragment should be given. Also, the primers used for cloning and the detailed transformation protocol should be reported.

Response: Details of the construction are now supplemented in the methods including the primers used in the study (Supplementary Table 1). We used the 5,353 bp fragment because the promoter of *SRKIR^{Col-0}* was located at upstream of the start codon. To support our idea that the continuous hairpin structure is required to suppress *SRKb*, we also created a transgenic plant with a genomic fragment containing only the front half of the inverted repeat. This new data is added as part of Fig. 2. Because this construct was unable to suppress the expression of *SRKb*, we consider intact *SRKIR^{Col-0}* is the suppressor.

- [M] The protocol to evaluate transcript abundance is not even mentioned and should be fully detailed. Where along the gene were the qPCR primers designed and how were they tested and optimized ? How many biological/technical replicates ? A specific concern is whether the primers are specific across the different SRK alleles. How have the authors validated their specificity ?

Response: We agree that previous version lacked the information of primers used in this study. They are now summarized in Supplementary Table 1. Numbers of biological replicates and technical replicates are written in figure captions. Primers are designed on sequences specific to each *SRKs* and they are not expected to hybridize across different alleles (excluding native *SRKb* and *SynSRKb* that were amplified by the same set of primers). Specificity of the amplifications were confirmed by melting curve analyses.

- [M] Along the same line, the statistical method to compare transcript abundance in Fig. 3d and 3e is not mentioned. As I understand, the authors are just doing visual comparison, but the fact that the wildtype ecotypes are homozygotes while the hybrids are heterozygotes renders the comparison far from obvious (a ratio of 1/2 should be expected even in the absence of any suppression, and should be used as the null hypothesis). It is essential that the statistical method is improved. Also, the label of the Y axis is unclear: does that correspond to ratios of Ct values ? In the legend, the word "expression" is not correct : what is measured is transcript levels. In Fig 3b I think the X axis is mislabelled. The second column should read C24/Sb and the third Col-0/Sb.

Response: We did the statistical analysis with the assumption that null hypothesis is 1/2 of the transcript accumulation levels in the homozygous wildtype strains. The results still shows that mRNA transcript accumulations are significantly reduced in the hybrids, both using Col-0 and C24 as parents.

Relative values were obtained by comparing transcript abundance of each *SRK* alleles against the endogenous *ACT8* gene. They were calculated from Cq values and PCR efficiencies, following the Roche protocol.

We apologize that C24/S_b and Col-0/S_a in Fig. 3b were misplaced in the previous version. It is now fixed.

- [m] The role of the C24 hairpin is not established as firmly as that the Col-0 hairpin, as not functional validation by genetic transformation was performed. Also, whether this other hairpin also produces small RNA was not tested. This should be straightforward to determine

since flower buds sRNA data are available from the literature (see e.g. <https://www.ncbi.nlm.nih.gov/geo/query/acc.cgi?acc=GSE85551>, although the exact stage might not be appropriate). At the very least this limitation should be clearly acknowledged.

Response: We are grateful to this very effective suggestion by the reviewer here. We indeed investigated this C24 flower sRNA data. The data clearly showed that significant number of reads map to the *SRKIR^{C24}* region, supporting our proposal. This data was added to Supplementary Figure 7.

- [M] The last paragraph mentions “the evolution of a broader epigenetic regulatory system” and links this to “the formation of micro RNA genes”. This wording is inappropriate on several grounds. One, at present there is no reason to call the suppression phenomenon “epigenetic”, as this is mostly presented as classical genetic regulation (PTGS) in the manuscript. Second, beside their size (20nt, which is unusual by the way), the link with microRNA genes is not clear. Canonical microRNA genes in plants are typically much shorter hairpin structures producing a largely unique (mature) 21nt molecule, which is very different from the pattern observed here. I would suggest to simply remove this sentence.

Response: We agree that this paragraph has small relevance to the main text. We removed the sentences.

- [m] The layouts of Fig 1a and Fig S1 are different. It would be a lot easier to follow if they were identical.

Response: We transformed the layouts of Supplementary Figure 1 to resemble Figure 1a.

- [M] In Figure 1a, an important control is missing, to show that pollen from Col-0 SRKb-SCRb is indeed fertile and able to germinate on stage 13 stigmas.

Response: We included this control to Figure 1a and 1b.

- [M] The method for the pollination experiments “other than Fig 1” is described in details, but nothing is reported for Fig. 1. Please clarify.

Response: Sorry for causing the confusion. This was a mis-statement. “Other than Fig 1” should have been “other than Supplementary Fig. 1”, because Supplementary Fig. 1 also used pistils from stage 13. Explanation here was revised.

- [m] Line 87-88, the stage at which the pollination experiments were performed should be given in the text, not just the figure legend.

Response: We revised the manuscript and the text now indicates the flower stage used (14) accordingly.

- [m] A consequence of the model proposed is that based on sequence information alone, one should be able to predict whether a given SRK allele can be successfully introduced into a given *A. thaliana* accession by transformation. This conclusion could perhaps be added to the discussion.

Response: We added a sentence to the discussion: “Based on these results, we consider that prediction of *SRK* alleles that could express its function when introduced into Col-0 or C24 is possible *in silico*.” based on the suggestion here by the reviewer.

Review by Vincent Castric

Reviewer #2 (Remarks to the Author):

This manuscript provides evidence suggesting that suppression of function of self-incompatibility in *Arabidopsis thaliana* can be driven by post-transcriptional gene silencing mediated through small RNAs. The manuscript is well-written, the results are interesting and the experiments conducted appropriately but I wasn't convinced that there is a substantive enough gain in knowledge to warrant publication in a high impact general journal. The short format also means that the story is not really told with sufficient background for a general audience to understand the context or the novelty and the methods are too brief to fully explain what was done.

I have made specific suggestions on an annotated version of the manuscript but my major concern is the presentation of the context and novelty, not only for a general audience but also for experts.

Response: We are grateful to this reviewer for giving detailed and insightful comments in the text itself. We have revised the manuscript accordingly. Based on the advices, we paid special attention to increase the accessibility from broad readership. We added a new paragraph introducing the evolution of SC and S haplotypes in *A. thaliana*. We also cited *SCR/SP11* dominance relationship papers in *Brassica* and *Arabidopsis* (*lyrata* and *halleri*) to introduce how SI studies contributed to understand genetic dominance. More explanations in other paragraphs were also added to improve the flow of the manuscript.

1) The presentation of background literature is biased to a few studies from a few laboratory groups, rather than presenting the broader body of literature suggesting possible mechanisms for loss of function of self-incompatibility in *A. thaliana* and its close relatives (including the extensive body of work on Brassicas).

Response: We agree that more detailed background could help this manuscript put in broader context. In correspondence to this comment, we introduced a paragraph dedicated for explaining the evolutionary loss of SI in *A. thaliana*. Specifically, relationship between polymorphic nature of SI and evolution of dominance, evolution of SC and S haplotypes in *A. thaliana* are now introduced to increase the connectivity.

2) There is also not enough context set about the background within *A. thaliana* in the introduction; more details about the variation in SI alleles is presented in the methods and results but it would help the reader and be more logical to provide all of the background to motivate the study in the introduction.

Response: Similar comment was given from the reviewer 1 and we agree that more detailed background on S haplotypes in *A. thaliana* is necessary. As suggested, Wei-1/Uk-3 belong to haplogroup A, Cvi-0 to B, C24 to R2 is now reported in the introduction of the current version.

3) The reader is also not told until the discussion that the inversions that are the focus of the paper had already been described and their frequency across accessions of *A. thaliana* quantified in a previous study. This compromises the novelty of the paper as currently presented. However, with refocusing, adding value to the previous study by the experimental testing in this study could broaden the impact of the paper.

Response: We agree that explanation on this part was not explicit. In this version we straightened out what has been mentioned in Tsuchimatsu et al MBE 2017 and what is found from our present study in the discussion. In short, Tsuchimatsu et al MBE 2017 has focused on the classification of the haplotypes and investigated how each haplotypes are geographically spread. They did not mention on the inverted repeats and that they are dominant.

In this paper, we showed that at least *SRKIR*^{Col-0} is functional, and *SRKIR*^{C24} is also likely to have a suppressing function against complementary *SRK* sequences. Taking account the work by Tsuchimatsu et al MBE, we predicted that they are dominantly spread in the species *Arabidopsis thaliana*. We believe, as mentioned by the reviewer here, our present result is in line with the results presented by Tsuchimatsu et al, except that we add refocusing views that inverted repeats may have contributed to the spread of these alleles in *A. thaliana*.

4) The paper generally lacks a logical flow of information. Various hypotheses are suggested in the methods, without explaining the specific logic that motivated them; it would be clearer to motivate the hypotheses in the introduction and then use the methods and results to test them. I have noted specific examples on the annotated version of the manuscript. It is quite confusing whether the hypotheses emerged from this study or were motivated by previous results.

Response: We are grateful to these comments on the text itself. We believe that we made responsive corrections for all of the specific comments in the main text given by the reviewer. In this version we carefully separated hypotheses that came from previous study, from those newly raised in our present study.

5) There is effectively no description of the statistics and there is a lack of quantitative details throughout the results (e.g. "high" sequence similarity).

Response: There was a similar suggestion from reviewer 1 on using tools to predict small RNA targetability. We used one of the available web interfaces called psRNAtarget (Dai et al 2018). The result was consistent with our prediction that *SRKIR^{Col-0}* produce small RNAs that could possibly target all *SRKb*, *SRK^{Wei-1}* and *SRK^{Col-0}*, while *SRKIR^{C24}* may not produce those that suppress *SRKb*. Statements related to the specificity of the *SRKIRs* is mainly based on the psRNAtarget predictions (the measure which is presumably more objective than the sequence similarities) in the current version.

6) I do appreciate that the short format means that describing some methods in the results is necessary but the results are dominated by methods and the methods are too brief to allow repeatability.

Response: This point was also commented by reviewer 1. In the current version, we put much more details (i.e. transgenic experiments, genotyping, RNA studies, bioinformatic analysis, statistical analysis) to the methods.

Reviewer #3 (Remarks to the Author):

This study by Fujii et al., 'Parallel evolution of dominant pistil-side self-incompatibility suppressors in Arabidopsis' identifies inverted repeat sequences in different regions of relic or pseudo SRK genes in Col-0 and C-24 respectively, that are responsible for silencing SRK genes through PTGS. They propose that these evolved in a parallel fashion and in populations where SC is favoured these mutations may have spread faster than a recessive mutation in SRK.

The work primarily focused on the discrepancy of why SI response in Col-0 was transient while in C24 it was more stable. The authors show that SRKb is efficiently targeted by the 502 bp inverted repeat corresponding to the kinase domain of SRKb. They also find a second mutation in C24, a 434-bp putative inverted repeat sequence within the pseudo SRK region of C24 (SRKIRc24) and this has low similarity with SRKb thereby, not affecting SRKb expression.

Although the idea of inverted repeats leading to silencing is not novel, this is the first report explaining why Col-0 plants show transient SI when SRKb/SCRb are expressed. While overexpression and expressing a synthetic version are quite promising, the ultimate proof would be to eliminate this 502 bp repeat sequence to show that a strong SI response can be conferred by expression of the SRKb/SCRb combo. The authors should genome edit Col-0 to eliminate the 502 bp region to show that SRKb can be more stable and manifest a strong SI response in the gene-edited (502 bp removed) background.

Response: It is true that elimination of 502 bp region can directly support our hypothesis. Instead, we included our new data using the line with T-DNA inserted in the loop region of *SRKIR^{Col-0}* that we had in our hands. As expected, transcript accumulation of *SRKb* mRNA in this T-DNA insertion line (with *SRKb* introduced) was restored compared to those that do not carry T-DNA (Supplementary Fig. 5). We also have additional data introducing the partial *SRKIR^{Col-0}* without the second inverted repeat (Figure 2). This partial *SRKIR^{Col-0}* failed to silence the function of *SRKb*, when the full *SRKIR^{Col-0}* was able to do so. Therefore, we consider that the continuous inverted repeat structure of *SRKIR^{Col-0}* is responsible for suppression of other *SRK* genes.

Other minor points:

1. Supplementary Fig. 1, The authors say they generated these lines (Col-0 + SRKb-SCRb). Are these pre-existing lines or newly created transgenic lines?

Response: This is the line generated by our group for this study. The genetic construct was generated in this study. We describe these details of the transgenic experiments in the methods of this version.

2. Figure 2, What promoter was used when introducing SRKIRcol-0 in to C24 background? Were any other SRK-related or S-domain kinases suppressed? Please show suppression of SRKb in these transgenic lines.

Response: Native promoter of the *SRKIR^{Col-0}* gene (AT4g21370) was used. We added the real-time RT-PCR data of *SRKb* expression in the *SRKIR^{Col-0}* introduced line (Figure 2d). In the same context we added the data of transgenic line introduced with genomic fragment carrying only the front repeat

of the $SRKIR^{Col-0}$ (partial $SRKIR^{Col-0}$). The gene expression of $SRKb$ was significantly suppressed in the $SRKIR^{Col-0}$ introduced lines, when it was not suppressed in the line introduced with partial $SRKIR^{Col-0}$.

3. Figure 3, while SCRb is dominant over SCRa, the SRKIRcol-0 inverted repeat should target both SRKa and SRKb in these lines. What is the expression level of SRKa in these lines (Col-0 and C-24 expressing Sa) in spite of being compatible with pollen from A. lyrata SaSb. We would expect suppression of SRKa in Col-0/Sa line.

Response: As we did not have any line that generate pollen expressing SCRa-specificity (neither natural *A. lyrata* line or transgenic *A. thaliana* line), we did not make an effort to investigate $SRKa$ mRNA expression. $SRKa$ does have a homology with $SRKIR^{Col-0}$ so we agree with the reviewer that we should see a suppression of $SRKa$ in the Col-0/Sa line. According to psRNAtarget, 80 non-redundant small RNAs perfectly match against $SRKa$.

Suppression of $SRKa$ by $SRKIR^{Col-0}$ is now under investigation. However, flowering of these inter-specific hybrids are not so uniform and would take much more time to perform experiments.

4. Figure 4, what promoter was used to express the synthetic SRKb gene in Col-0 background? As a control native gene under the same promoter should be expressed in Col-0 background to examine if it can be suppressed by SRKIRcol-0 and a compromise in SI should be observed.

Response: Both native and synthetic versions of $SRKb$ were controlled by the same promoter, the pseudo $SRKA$ gene in Col-0 (AT4g21370). $SRKb$ moiety of the $SRKb$ - $SCRb$ /pBI121 construct was replaced by *synSRKb*. This is now described in the methods in detail.

5. The authors should discuss the parallel evolution, is it parallel with the inverted repeat identified in the pollen SCR fragments or between the inverted repeats of SRK in Col-0 and C-24

Response: This is made clear in the current version. Inversion of SCR fragments is considered to have occurred at the early stage of self-compatibility transition (Tsuchimatsu et al 2010). This is because some haplotype A strains retain functional SRK . We propose the parallel evolution of the inverted repeats $SRKIR^{Col-0}$ and $SRKIR^{C24}$, because they are independent sequence that evolved from the same ancestral $SRKA$.

REVIEWERS' COMMENTS:

Reviewer #1 (Remarks to the Author):

This new version of NCOMMS-19-26822A addresses most of my comments. The methods are now better described, the biological system is presented in more details and the new analyses (e.g. target predictions, new sRNA data in C24, new controls) confirmed the initial findings.

I still have two important points :

1. One is that the study is presented as a contribution to the debate on the evolution of dominance, while I think it does not do justice to the results presented and is somewhat misleading. The study does provide important insight on why reconstituting SI in *A. thaliana* has been straightforward in some ecotypes but not others, and that in itself is an important contribution. The fact that the mutations involved are dominant is interesting as "yet another" example of dominant-negative mutations, but it is somewhat peripheral, and has little to do with the processes by which the dominance hierarchy documented at the *S*-locus in allogamous species evolved. I am afraid that linking the two processes, as the manuscript currently does, will confuse the readers on a topic about which a lot of confusion has already been conveyed. I strongly recommend to focus in the introduction on the evolutionary history of the SI system in *A. thaliana*, and to keep the broader context of dominance for the discussion. As I explained in my previous review, dominant-negative mutations are not a new finding but there are not so many cases where they have been understood at the molecular level as well as in the present case, so it would be more appropriate to briefly mention this point at the end of the manuscript rather than to open it at length on these aspects.

2. The other is that I remain puzzled by the different conclusions between Liu et al. (2007) and this study on whether the SI modifier in Col-0 is dominant or recessive. The authors should comment on why this is so, and whether/if these two opposite conclusions can be reconciled.

Also:

- line 271, the mention to "climatic situations such as the glacial periods" needs some explanation.
- line 280 : still not clear why the process by which the inverted repeats silence the SRKs is referred to as "epigenetic" : this is good old genetics, no ?

Vincent Castric

Reviewer #2 (Remarks to the Author):

The authors have generally done a good job of addressing the comments raised by myself and other reviewers. The context is now set more appropriately and the methods are described in more detail.

The only comment that I raised that was not specifically addressed was in relation to the statistics. This is still all that is said about statistical analyses: "All of the statistical analyses and bar plot visualizations were carried out using R47)". This does not tell the reader what comparisons were made and using which statistical tests.

Barbara Mable

Reviewer #3 (Remarks to the Author):

Through supplementary Fig.5 using T-DNA insertion in the repeat region of SRKIRCol-0 in Col-0 background and expression of the partial repeat of SRKIRCol-0 in SI-C24 background, the authors have convincingly addressed my major concern. The added experiments along with expression data have considerably improved the manuscript.

To make this complete:

1. Please add aniline blue images for Fig.5
2. Add information on how many lines were isolated in SI-C24 background expressing the partial SRKIRCol-0 repeat and did they all manifest SI?

Reviewer #1 (Remarks to the Author):

This new version of NCOMMS-19-26822A addresses most of my comments. The methods are now better described, the biological system is presented in more details and the new analyses (e.g. target predictions, new sRNA data in C24, new controls) confirmed the initial findings.

I still have two important points :

1. One is that the study is presented as a contribution to the debate on the evolution of dominance, while I think it does not do justice to the results presented and is somewhat misleading. The study does provide important insight on why reconstituting SI in *A. thaliana* has been straightforward in some ecotypes but not others, and that in itself is an important contribution. The fact that the mutations involved are dominant is interesting as "yet another" example of dominant-negative mutations, but it is somewhat peripheral, and has little to do with the processes by which the dominance hierarchy documented at the S-locus in allogamous species evolved. I am afraid that linking the two processes, as the manuscript currently does, will confuse the readers on a topic about which a lot of confusion has already been conveyed. I strongly recommend to focus in the introduction on the evolutionary history of the SI system in *A. thaliana*, and to keep the broader context of dominance for the discussion. As I explained in my previous review, dominant-negative mutations are not a new finding but there are not so many cases where they have been understood at the molecular level as well as in the present case, so it would be more appropriate to briefly mention this point at the end of the manuscript rather than to open it at length on these aspects.

Response: We revised manuscript based on this comment. As also requested by the editor, we removed first two sentences and the last sentence of the abstract, and also the first paragraph of the introduction related to evolution of dominance. Abstract and introductions therefore starts with evolution of selfing and self-incompatibility. Statements related to the evolution of dominance is limited to the last part of discussion.

2. The other is that I remain puzzled by the different conclusions between Liu et al. (2007) and this study on whether the SI modifier in Col-0 is dominant or recessive. The authors should comment on why this is so, and whether/if these two opposite conclusions can be reconciled.

Response: We are also puzzled why Liu et al ended up to conclude that SI modifier factor in Col-0 is recessive. All of our experiments supported that the factor in Col-0 is a dominant suppressor: SI phenotype and SRK transcript suppression in the Col-0 x C24 hybrids (Figure 1), introduction of *SRKIR^{Col-0}* into SI-C24 compromises the SI phenotype (Figure 2), the Col-0 S allele suppresses *SRK* of other haplotypes including *A. lyrata* (Figure 3), and the effect of *SRKIR^{Col-0}* could be avoided by synonymous codon substitution of *SRKb* (Figure 4). In the first section of the results, we clearly indicate that the two studies disagree, and we believe this is all we can suggest at this moment.

Also:

- line 271, the mention to "climatic situations such as the glacial periods" needs some explanation.

Response: We added a phrase "when pollen availability is limited³¹" after this to be more specific.

- line 280 : still not clear why the process by which the inverted repeats silence the SRKs is referred to as "epigenetic" : this is good old genetics, no ?

Response: As we agreed that the molecular mechanism is still not clear, we removed the "epigenetic" mention.

Reviewer #2 (Remarks to the Author):

The authors have generally done a good job of addressing the comments raised by myself and other reviewers. The context is now set more appropriately and the methods are described in more detail.

The only comment that I raised that was not specifically addressed was in relation to the statistics. This is still all that is said about statistical analyses: "All of the statistical analyses and bar plot visualizations were carried out using R47)". This does not tell the reader what comparisons were made and using which statistical tests.

Response: We added the sentence "Dunnnett's test was performed using the *glht* function from the *multcomp* package⁴⁴." to address the package used for the statistical analysis in this study. Details of the statistics are given in each Figure captions.

Reviewer #3 (Remarks to the Author):

Through supplementary Fig.5 using T-DNA insertion in the repeat region of *SRKIRCol-0* in Col-0 background and expression of the partial repeat of *SRKIRCol-0* in SI-C24 background, the authors have convincingly addressed my major concern. The added experiments along with expression data have considerably improved the manuscript.

To make this complete:

1. Please add aniline blue images for Fig.5

Response: We added the image to Supplementary Fig. 5 as requested.

2. Add information on how many lines were isolated in SI-C24 background expressing the partial *SRKIRCol-0* repeat and did they all manifest SI?

Response: We isolated five independent T₁ lines and they all retained full SI ability comparable to SI-C24. We therefore chose to analyze only one line out of them and analyzed its *SRKb* transcript accumulation. This is stated in the methods now.